# Monte Carlo study of the pseudogap and superconductivity emerging from quantum magnetic fluctuations

Weilun Jiang[1,2], Yuzhi Liu [1,2], Avraham Klein [3], Yuxuan Wang[4], Kai Sun [5], Andrey V. Chubukov [6] &
Zi Yang Meng [1,7 ✉]

The origin of the pseudogap behavior, found in many high-$T_c$ superconductors, remains one
of the greatest puzzles in condensed matter physics. One possible mechanism is fermionic
incoherence, which near a quantum critical point allows pair formation but suppresses
superconductivity. Employing quantum Monte Carlo simulations of a model of itinerant
fermions coupled to ferromagnetic spin fluctuations, represented by a quantum rotor, we
report numerical evidence of pseudogap behavior, emerging from pairing fluctuations in a
quantum-critical non-Fermi liquid. Specifically, we observe enhanced pairing fluctuations and
a partial gap opening in the fermionic spectrum. However, the system remains non-
superconducting until reaching a much lower temperature. In the pseudogap regime the
system displays a "gap-filling" rather than "gap-closing" behavior, similar to the one observed
in cuprate superconductors. Our results present direct evidence of the pseudogap state,
driven by superconducting fluctuations.

---

[1] Beijing National Laboratory for Condensed Matter Physics and Institute of Physics, Chinese Academy of Sciences, Beijing 100190, China. [2] School of Physical
Sciences, University of Chinese Academy of Sciences, Beijing 100190, China. [3] Department of Physics, Faculty of Natural Sciences, Ariel University,
Ariel, Israel. [4] Department of Physics, University of Florida, Gainesville, FL 32601, USA. [5] Department of Physics, University of Michigan, Ann Arbor, MI
48109, USA. [6] School of Physics and Astronomy, University of Minnesota, Minneapolis, MN 55455, USA. [7] Department of Physics and HKU-UCAS Joint
Institute of Theoretical and Computational Physics, The University of Hong Kong, Pokfulam Road, Hong Kong SAR, China. ✉email: zymeng@hku.hk

Even though unconventional and high-$T_c$ superconductivity arises in a diverse set of materials, many of them share similar features in their phase diagram. One prominent feature is a superconducting (SC) dome, which emerges near the termination point of a non-SC phase with either spin or charge order. The second feature is anomalous transport and non Fermi-liquid (nFL) behavior around the putative quantum critical points (QCP). These features have led to the proposal that soft quantum-critical fluctuations of the order parameter serve as the source for the universal behavior and mediate singular interaction that gives rise to superconductivity with nontrivial pairing symmetry, strange metal behavior, and intertwined orders.

In many unconventional superconductors, most notably the cuprates, there is a third feature: the "pseudogap(PG)", a gap-like feature in the fermionic spectrum above the SC phase. Despite decades of investigation, the origin (or origins) of the PG remain intensely debated. One class of proposals names exotic, possibly topological order in the particle-hole channel as the origin[1–3], while another points to pairing fluctuations in the strong coupling regime[4–10]. Substantial numerical efforts have been dedicated to the understanding of PG, see e.g., refs. [1,11,12] and references therein.

The understanding of the coupling between fermionic excitations near the Fermi surface (FS) and bosonic quantum critical fluctuations[14–18] is crucial to describe these three features. The development of quantum Monte Carlo (QMC) algorithms for a class of models of this type, pioneered by ref. [19], has created a feasible way to study this physics in an unbiased manner (see the reviews[20,21] and references within). In QMC models, FS fermions couple to bosonic fluctuations, representing certain collective modes of low-energy fermions[22–29]. The bosonic part is bestowed with independent (non-fermionic) dynamics and can be tuned to criticality to mimic the situation in real materials. Crucially, these models are free of the sign-problem plaguing most fermionic QMC, allowing for a realistic test of theory.

In this work, we investigate the PG physics via such a sign-free QMC simulation of fermions near a ferromagnetic QCP. We find robust signatures of a PG above the SC state and are able to study its spectral properties and its interplay with the dynamics of the ferromagnetic degrees of freedom. We also compare the numerical results with several theoretical predictions, and reconcile many key aspects of the two.

## Results

**Overview**. Before going into the details of our work, we present an overview of the essential features of our model and a summary of the main results.

The model we choose to implement is a variant of a quantum critical model, in which the bosons represent critical ferromagnetic(FM) spin fluctuations (a "spin-fermion" model). When looking for a spectral property of the superconductivity, such as a PG, such a model has an advantage over analogous ones, e.g., antiferromagnetic or nematic models (see e.g.,[21]) because of the simplicity of the momentum structure of the FS (e.g., no hot or cold spots). Furthermore, compared to earlier sign-free QMC studies on ferromagnetism, the coupling strength of our model is stronger in two aspects. First, the spin system is an XY quantum rotor model that is inherently more strongly fluctuating than an Ising model, analyzed earlier[30,31]. Second, the coupling constant $K$ between the fermionic and bosonic sectors is set to larger values than in previous works. The larger coupling pushes the region of SC fluctuations up to temperatures, where they are discernible in the numerical data. This in contrast with earlier works, where coupling strength was optimized to study normal state properties.

As we see below, the larger coupling allows us to reveal the PG behavior.

In the normal state, at low enough temperatures we find in the bosonic sector near the QCP an overdamped dynamics with linear frequency response ($z = 2$ scaling). This is different from the $z = 3$ behavior, found in Ising systems, and is a result of a non-conservation of the order parameter in our model.

In the temperature range, where the bosonic susceptibility is linear in frequency, we observe several remarkable features. The uniform susceptibility deviates from Curie-Weiss behavior and actually becomes weaker at smaller $T$. In the fermionic sector, we find a gap-like feature in the density of states (DOS). Unlike in a BCS superconductor, the size of the gap remains roughly independent on temperature, while the DOS becomes progressively depleted (filled) upon lowering (raising) temperature. Importantly, the scaling behavior of the pairing susceptibility clearly shows that the system is not in a SC state. We thus identify the spectral gap in such a state as a PG.

We note that the "gap-filling" behavior observed in our numerical results has also been observed in tunneling and photoemission experiments on the cuprates[32], and has been obtained in a class of $\gamma$ − models of quantum-critical pairing [6]. Our results, obtained from unbiased large-scale QMC simulations, confirm the existence of a PG behavior from pairing fluctuations in a quantum-critical system with itinerant fermions.

The quantum-critical spin dynamics and normal state fermionic properties that we find are consistent with recent theoretical predictions for nFLs at finite temperature, obtained within the modified Eliashberg theory[31,33,34]. This allows us to benchmark our simulations and extract relevant parameters from the observables (see Supplementary Note 4 for details). The onset temperature for PG behavior, $T_{PG}$, and SC $T_c$, extracted from QMC, are consistent with theoretical predictions (see ref. [6] and Methods). Our results therefore provide an attempt to numerically realize the transition from nFL to PG and eventually to superconductivity, lending support to the scenario of pairing fluctuations driven PG phenomena.

Crucially, we view our finding of a PG as the evidence of a universal mechanism for the formation of a PG from SC fluctuations near a QCP, not limited to the specific model of FM spin fluctuations that we used. We do not claim that we present a model for PG formation in a specific material, but in view of our findings we do expect SC fluctuations to be a contributing factor to PG formation in any system close enough to a QCP, independently of the specific origin of the pairing boson.

The PG, obtained in our work, comes from pairing fluctuations in a situation when the pairing is in turn mediated by a propagator of a FM order parameter. While our model does not directly describe experimental situation in the cuprates, where antiferromagnetic fluctuations are often considered to be a pairing glue, we argue that the mechanism for the PG formation, studied in our work, is a universal phenomenon of the pairing near a quantum-critical point[6], and in this sense goes beyond the specific model with FM fluctuations. We do expect the SC fluctuations to be a contributing factor to PG formation in any system close enough to a QCP, independently of the specific origin of the pairing glue.

## Model

We consider a model of itinerant fermions coupled to SO(2) quantum rotors, as shown in Fig. 1a (rotors are in the middle layer). The model is described by

$$\hat{H} = \hat{H}_{qr} + \hat{H}_f + \hat{H}_{qr-f}, \tag{1}$$

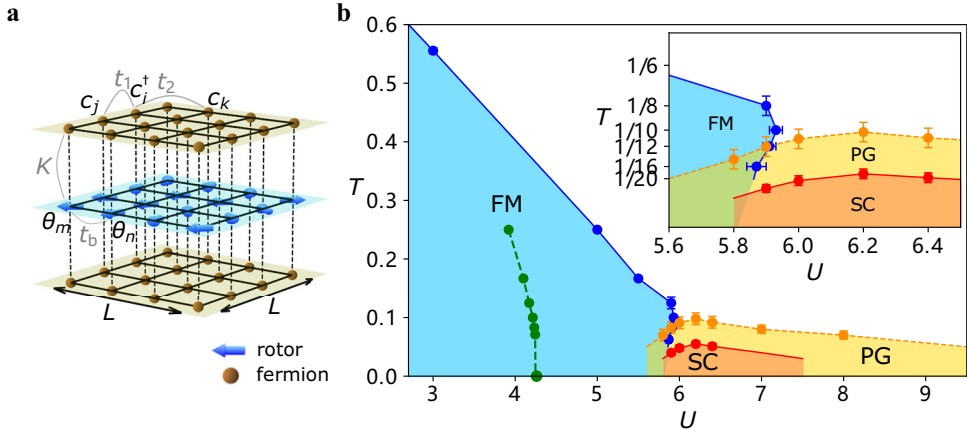

**Fig. 1 Model and Phase diagram. a** Sketch of the model in Eq. (1). Deep yellow dots and grids of the top and bottom layers represent fermion degrees of freedom with nearest hopping strength $t_1$ (e.g., $\hat{c}_i^\dagger \hat{c}_j$) and next-nearest hopping $t_2$ (e.g., $\hat{c}_i^\dagger \hat{c}_k$). Blue arrows and grid in the middle layers denote bosonic parts with an unit vector representing $\theta \in [0, 2\pi)$ of the rotor on each site. The interaction between two rotors on nearest sites (e.g., $\theta_m$ and $\theta_n$) is $t_b$. The on-site coupling $K$ between fermions and bosons is shown by the vertical dashed lines. The system size is $L \times L$. **b** $U - T$ phase diagram of the model obtained from QMC simulation. The inset zooms in to the vicinity of the pseudogap (PG, yellow), ferromagnetic (FM, blue) and superconducting (SC, orange) regions. The blue points on the FM phase boundary are determined by finite size scaling with fixed $T$ or $U$. Notably, for $U = 5.9$, as temperature gets lower, the system first enters into the FM phase at $T \approx 0.13$, then exits it at $T \approx 0.08$. The yellow points of the PG boundary are determined from the onset of a PG in the single-particle spectrum, as shown in Fig. 2. The red points denoting an onset of s-wave superconductivity are determined from the onset of a full gap in the spectrum as well as Kosterlitz-Thouless scaling of the pairing susceptibility. The maximum of SC phase transition temperature $T_c$ is ~0.05. The green points and dashed line, are the phase boundary of the (uncoupled) quantum rotor model[13]. See the Supplementary Note 3 for additional details as well as a discussion of SC fluctuations above $T_c$. The errorbars of the points on the FM phase and SC phase boundaries are determined by the data collapse with fixed $T$ or $U$. For the PG phase boundary, the errorbars come from the uncertainty in identifying the onset of the minimum at $\omega = 0$ for distinct temperatures of DOS.

where

$$\hat{H}_{qr} = \frac{U}{2} \sum_i \hat{L}_i^2 - t_b \sum_{\langle i,j \rangle} \cos\left(\hat{\theta}_i - \hat{\theta}_j\right)$$

$$\hat{H}_f = -t_1 \sum_{\langle i,j \rangle \sigma \lambda} \hat{c}_{i\sigma\lambda}^\dagger \hat{c}_{j\sigma\lambda} - t_2 \sum_{\langle\langle i,j \rangle\rangle \sigma \lambda} \hat{c}_{i\sigma\lambda}^\dagger \hat{c}_{j\sigma\lambda} - \mu \sum_{i\sigma\lambda} \hat{n}_{i\sigma\lambda} \quad (2)$$

$$\hat{H}_{qr-f} = -\frac{K}{2} \sum_{i\lambda} \left(\hat{c}_{i\lambda}^\dagger \sigma^x \hat{c}_{i\lambda} \cdot \cos\hat{\theta}_i + \hat{c}_{i\lambda}^\dagger \sigma^y \hat{c}_{i\lambda} \cdot \sin\hat{\theta}_i\right).$$

The first term $\hat{H}_{qr}$ describes a quantum rotor model on a square lattice. Here $\hat{L}_i$ is the angular momentum of 2D rotor $\hat{\theta}_i$ at site $i$. The second term $\hat{H}_f$ describes two identical copies of spin-1/2 fermions on a square lattice, with layer index $\lambda = 1$ and $2$ representing the top and the bottom layers. Fermions in each layer can hop between nearest-neighbor (next-nearest-neighbor) sites with hopping amplitudes $t_1$ ($t_2$), and the chemical potential $\mu$ controls the fermion density. The last term $\hat{H}_{qr-f}$ couples quantum rotors and fermions via an on-site FM interaction that tends to align XY component of a fermion spin with the direction of a rotor on each site.

In the absence of fermion-rotor coupling, rotors develop quasi-long-range FM order via a Kosterlitz-Thouless(KT) transition[13,35]. At zero temperature, FM order becomes long range. The KT transition line in $(T, U)$ plane terminates at a QCP at $(U/t_b)_c = 4.25(2)$[13,36,37]. As we turn on the fermion-rotor coupling, fermion contributions shift the KT phase boundary towards larger $U$ and $T$. More importantly, the phase transition now involves fermion spins, which at $T = 0$ also order ferromagnetically below $U_c$. This allows us to study quantum phenomena near a FM QCP in a metal[38]. Due to the anti-unitary symmetry and the presence of two copies of fermions, this model can be simulated via QMC techniques without the sign problem (see Supplementary Note 1 for details). This setup then allows us to analyze the universal behavior near a QCP with high numerical accuracy and large system sizes.

We express all quantities in units of $t_b$. In the simulations we set $K = 4$, $t_1 = 1$, $t_2 = 0.2$ and $\mu = 0$. We varied $U$ and the temperature $T$ and constructed the phase diagram of the model, Fig. 1b, which features a paramagnetic-ferromagnetic transition and several other transitions/phases. The magnetic transition at a finite temperature is of KT type. As $U$ increases, the transition temperature decreases and terminates at a QCP at $U_c$. The $T = 0$ transition upon varying $U$ belongs to XY universality class as the coupling to rotors creates an easy plane for fermion spins. Fermion spins order ferromagnetically in the XY plane, breaking a spin-rotational symmetry.

**Pseudogap and superconductivity properties.** We observe a SC dome around the QCP. Above the dome, we find evidence of PG behavior in the range of $T$, whose width is comparable to $T_c$.

First, by measuring correlation functions of Cooper pairs in various pairing channels, we found that the dominant pairing channel is spin-triplet and odd under the interchange between the top and the bottom layers (layer-singlet), i.e., $\Delta(\mathbf{r}) = \frac{1}{\sqrt{2}}(\hat{c}_{\mathbf{r}1\uparrow}\hat{c}_{\mathbf{r}2\downarrow} - \hat{c}_{\mathbf{r}2\uparrow}\hat{c}_{\mathbf{r}1\downarrow}) = \frac{1}{\sqrt{2}}(\hat{c}_{\mathbf{r}1\uparrow}\hat{c}_{\mathbf{r}2\downarrow} + \hat{c}_{\mathbf{r}1\downarrow}\hat{c}_{\mathbf{r}2\uparrow})$, where 1 and 2 label layers. In the classification of 2D irreducible representations, this is an s-wave gap, as $\Delta(0)$ is finite. We verified (see Supplementary Figs. 1 and 2) that the susceptibility in this channel strongly increases when the system approaches a superconducting instability, while the susceptibilities in all other pairing channels remain small and do not increase. This observation is a direct evidence that superconductivity originates from the interaction mediated by soft bosonic fluctuations, associated with the QCP. Indeed, it has long being known that near a FM quantum phase transition, soft dynamical bosonic fluctuations introduce an effective interaction that is attractive in the spin-triplet channel. In the geometry of our model, there are two distinct types of spin-triplet pairing—one is odd under momentum inversion in a layer and even under layer interchange (e.g., p-wave layer-triplet), the other is even within each layer and

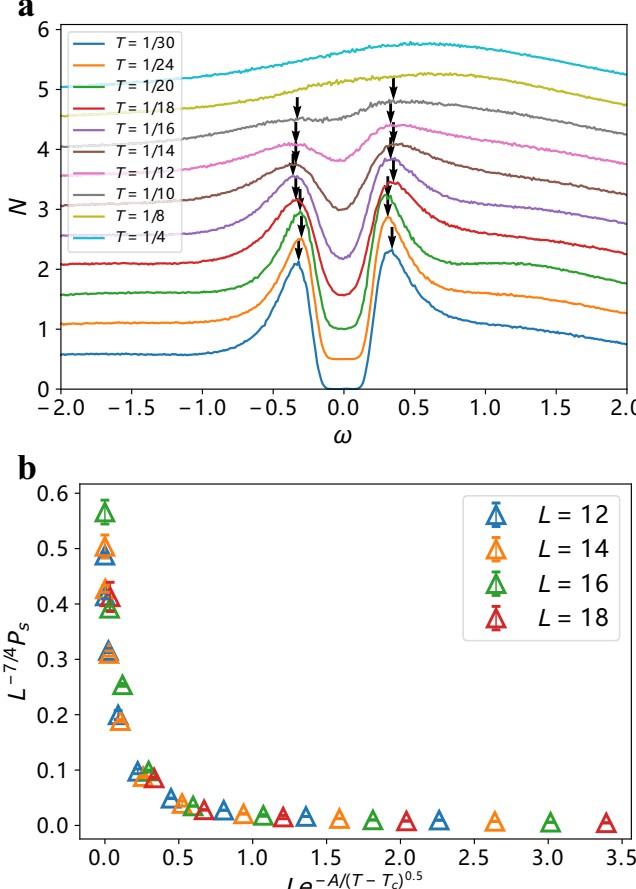

**Fig. 2 Pseudogap and superconductivity. a** Local DOS $N(\omega)$ for various temperatures at $U = 6$ with $L = 12$. For $T = 1/4$, far above the PG, the system exhibits a Fermi liquid spectrum. At $T_{PG} \approx 0.1$, the SC fluctuations begin to play important role and a noticable gap forms at $\omega = 0$. This gap-forming temperature is consistent with the corresponding intermediate temperature scale in the dynamic bosonic susceptibility $\chi$ in Fig. 4b. The gap minimum $N(\omega = 0)$ goes down with temperature and eventually reaches zero at $T \approx 0.05$, indicating the onset of a SC state as detected by the pairing susceptibility in **b**. At $T < 0.05$, fully gapped spectrum corresponds to the behavior of the SC state. **b** Data collapse of the pairing susceptibility $P_s$ versus temperature at $U = 6$ for system sizes $L = 12, 14, 16, 18$ with statistical errors obtained by QMC simulations, consistent with a KT transition. The best fit coefficients are $A = 0.75$, $T_c = 0.048$, which is consistent with the temperature of the fully-gapped spectrum in **a**.

odd under layer interchange (s-wave layer-singlet). By analogy with previous studies of the pairing mediated by small **q** fluctuations[30], one expects the leading instability to be towards the s-wave layer-singlet, spin-triplet order. The numerical finding of the largest pairing correlations in this channel thus affirms the crucial role of soft FM bosonic fluctuations in the formation of a SC dome.

Second, we obtained the fermionic spectral function, and then integrated it over $k - $ space to obtain the DOS $N(\omega)$. For this, we first computed the imaginary-time fermion Green's function and then converted it to real frequency via stochastic analytic continuation (SAC) method (See Methods for details). We show the results for $N(\omega)$ in Fig. 2a. At low $T$, inside the SC dome, there is clear evidence for an s-wave gap. The data shows that, that as $T$ increases, the magnitude of the gap slightly increases, rather than shrinks, as would be the case in a BCS superconductor. Simultaneously, $N(\omega)$ for $\omega$ smaller than the gap increases and

gradually fills in the states within the gap, ultimately restoring its normal-state value. This phenomenon has been termed gap-filling. It is qualitatively in agreement with experimental observations in many strongly-correlated unconventional super-conductors at $T \geq T_c$[5,9,10,39], At smaller $T \leq T_c$, the DOS displays gap-closing behavior, like in a conventional BCS superconductor. Guided by the experimental evidence[9,10] that gap-filling behavior holds at $T \geq T_c$, we defined the PG region as the one where the DOS gets filled in upon increasing $T$. We set the lower boundary of this region to where the DOS at the Fermi energy significantly deviates from thermally activated behavior of $e^{-\Delta/k_B T}$. The upper boundary of the PG region is set at $T_{PG}$, at which the dip of $N(\omega)$ at the Fermi energy becomes invisible. The PG region, obtained this way, is plotted in yellow in Fig. 1b.

Third, to determine the actual SC transition temperature, $T_c$, we performed scaling analysis of the pairing susceptibility $P_s = \frac{1}{L^2} \int_0^\beta \sum_i \langle \Delta^\dagger(\mathbf{r}_i, \tau) \Delta(\mathbf{0}, 0) \rangle$, using KT scaling for the pairing susceptibility $P_s = L^{2-\eta} f(L \cdot \exp(-\frac{A}{(T-T_c)^{1/2}}))$ for $T > T_c$ with $\eta_{KT} = 1/4$[28,40,41]. We show the results in Fig. 2b. The data for $P_s$ for various system sizes and temperatures collapse onto a single curve. We fitted the curve by the formula above and extracted $T_c = 0.048$. This agrees with the lower boundary of the PG region. The upper boundary, $T_{PG}$, is about twice larger in our simulations, $T_{PG} \sim 0.1$. We also computed the superfluid density, $\rho_s(T)$, which has been widely used to estimate $T_c$ in QMC simulations. This is done by detecting the temperature $T_\rho$ at which $\rho_s(T_\rho) = \alpha T_\rho$, where $\alpha$ is a dimensionless constant[41], usually set to $2/\pi$, based on the analysis of the XY model[42]. This criterion, although qualitatively correct, typically overestimates $T_c$[40]. In our case, we found $T_c < T_\rho \sim T_{PG}$. We discuss our analysis of $\rho_s$ in some length in Supplementary Note 3.

We analyzed the QMC data within the quantum critical theory of itinerant ferromagnets[43,44], extended to finite $T$[34] and modified to include two layers of itinerant fermions and superconductivity. We computed fermionic and bosonic self-energies near $U_c$ and found good agreement with the simulations in the normal state (see Supplementary Note 4). We extracted the effective fermion-boson coupling from this comparison, and used it to compute the onset temperature for the pairing within the Eliashberg theory for quantum-critical pairing[6]. This theory does not differentiate between pair formation and superconductivity, hence the result has to be compared with $T_{PG}$, extracted from simulations. We obtained theoretical $T_{PG} \sim 0.08$, quite consistent with $T_{PG} \sim 0.1$, extracted from QMC data, see Fig. 1b. Further, Eliashberg calculations below $T_{PG}$ show gap-closing behavior at small $T$ and gap-filling behavior at $T \leq T_{PG}$. The boundary between two regimes has been associated with the actual $T_c$, based on the analysis of phase fluctuations[6]. We show this in the phase diagram in Fig. 1b. Based on this comparison, we argue that our unbiased numerical QMC simulations are consistent with the theory and provide strong evidence for PG behavior, originating from preformed pairs above $T_c$, near a FM QCP in a metal.

We note that previous QMC work (see e.g.,[22,23] for an antiferromagnetic model) found an SC dome surrounded by a region of SC fluctuations. These were determined by comparing $T_\rho$ determined from the BKT criterion for $\rho_s$ (see above), and the temperature $T_{dia}$ at which the system showed diamagnetic behavior, as evidenced by a sign-change of the appropriate current-current correlator. While we too find such fluctuations (see Supplementary Note 3). We stress that the region of gap-filling behavior is predominantly at $T_c < T < T_{PG} \sim T_\rho$, and is therefore distinct from fluctuations on the scale of $T_{dia}$. Indeed, from our numerics we observe that $T_{PG} < T_{dia}$. We emphasize that in the thermodynamic limit, $T_{PG}$ and $T_{dia}$ do not correspond to phase transitions, but rather mark crossover regions.

**Magnetic dynamics and re-entrance effect.** The pairing behavior also has an impact on the magnetic phase transition and the quantum dynamics of the rotors. As shown in Fig. 1b, the phase boundary of the paramagnetic-ferromagnetic transition exhibits a re-entrance behavior at $U \sim 5.9$, close to the QCP. For example, at $U = 5.9$, upon reducing the temperature, the system first enters the FM state and then returns to the paramagnetic one, i.e., there is a "back-bending" of the transition line to the FM state. This can also be seen from the Fermi surface behavior. In Fig. 3, we plot the Fermi surface, $G(\mathbf{k}, \tau = \beta/2) \sim N(\mathbf{k}, \omega = 0)$, evolution with temperature. At intermediate temperature $T = 0.1$, the Fermi surface splits due to the ferromangetic order. However, the split vanishes both either increasing or lowering the temperature. We believe that the re-entrance phenomenon is a consequence of the PG and SC fluctuations, which suppress the fermion DOS and hence the electron-hole contribution to magnetic order[45,46]. Similar behavior has been seen previously in an antiferromagnetic model[22], but no PG was reported there. We emphasize that the paramagnetic-ferromagnetic phase boundary starts to bend to the left roughly at $T_{PG}$, which is well above the SC dome, indicating that SC fluctuations without phase coherence in the PG region are responsible for the magnetic dynamics.

We note, that in the absence of an SC dome, previous works (see e.g.,[38,44,47,48]) have shown that itinerant FM QCPs are unstable to a first-order transition driven by normal state magnetic fluctuations, which can cut off the FM phase at larger $U$'s, similar to the behavior seen in our simulations. In our results, we do not observe clear evidence of a first-order magnetic transition, and the correlation of the back-bending with $T_{PG}$ implies that for the parameters that we used the physics is driven chiefly by SC rather than magnetic fluctuations.

In addition, we measured the inverse dynamical bosonic susceptibility of the rotors across different regions of the phase diagram. Our results are summarized in Fig. 4, showing data for three representative $U$ at various temperatures. To study the dynamics, we subtract the static part of the inverse susceptibility and focus on the spin polarization $\chi^{-1}(\mathbf{q}, \omega) - \chi^{-1}(\mathbf{q} = 0, \omega = 0)$ (for details of what follows see Supplementary Note 2 and 3). Deep in the FM phase, Fig. 4a, we find an $\omega^2$ dependence (dynamical exponent $z = 1$). This is similar to that of the bare rotor model, and indicates that the fermionic contribution to the dynamics is negligible because of the spin gap. Similarly, deep in the Fermi liquid phase, Fig. 4c, we find an $\omega^2$ dependence, except at the lowest frequencies, which furthermore extrapolates to a nonzero value. The saturation is readily understood as resulting from the non-analyticity of the Lindhard function which implies $\chi^{-1}(\mathbf{q} = 0, \omega \to 0) \neq \chi^{-1}(\mathbf{q} \to 0, \omega = 0)$ at weak coupling. In the quantum-critical regime, Fig. 4b, we find a qualitatively different behavior indicating strong fermionic correlations.

First, at higher frequencies we find a linear frequency dependence ($z = 2$), which does not saturate to a finite value. This is surprising at first glance, since Landau damping for a ferromagnet has an $\omega/q$ form ($z = 3$) rather than linear $\omega$. We note that, in purely electronic models $\chi^{-1}(\mathbf{q}, \omega)$ is required to be non-analytic at any coupling strength due to spin conservation, and non-analytic behavior was seen previously in simulations of Ising-ferromagnets. However, in our simulations of the XY model, the order parameter is not conserved, leading to linear frequency dependence, in direct contrast to Ising model studied in refs.[30,31].

Second, at lower Matsubara frequencies accessible at lower temperatures, the $\omega^2$ behavior is again restored even in the quantum-critical region. As discussed above, this is again a direct result of the formation of a gap—this time the pseudogap, which depletes the low-energy fermion density of states and reduces the fermionic feedback on the bosons.

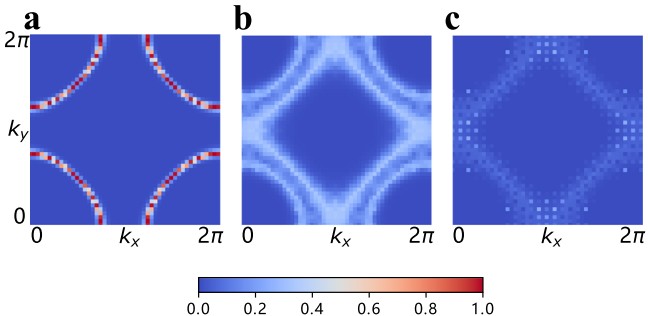

**Fig. 3 Re-entrance.** Evolution of the Fermi surface (FS) from non-interacting system with $H_f$ in **a**, to the nFL FS subjected to strong FM correlation at $U = 5.9$, $T = 0.1$ in **b**, and eventually to the FS in the PG phase at $U = 5.9$, $T = 0.05$ in **c**. The spectral weights are normalized with the same scale. The system size is $L = 12$ and the twisted boundary condition in the fermion hopping is applied such that the momentum resolution is 4 times larger in both $k_x$ and $k_y$ directions.

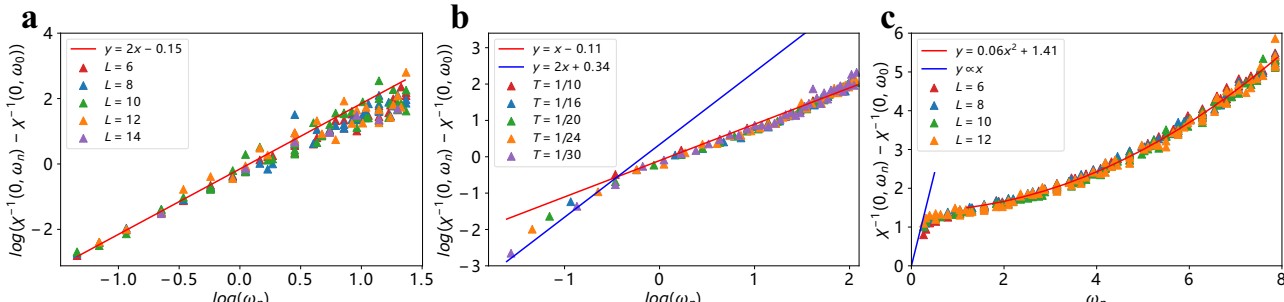

**Fig. 4 Magnetic dynamics.** Inverse bosonic dynamic susceptibility $\chi$ versus $\omega_n$ in three different regions, at $U = 3, 6, 8$, corresponding **a** in the FM phase, **b** in PG and SC phases, and **c** disordered phase. **a** log-log plot for various system size $L = 6, 8, 10, 12, 14$, each of which includes various $\beta = 12, 16, 20, 24$. Red line is a quadratic line of $\chi^{-1} \sim \omega^2$ for low frequency part $\omega_n < 1$. **b** log-log plot for various $\beta = 10, 16, 20, 24, 30$ with $L = 12$. At temperature $T = 0.1$ ($\beta = 10$), the fermions are in the quantum critical regime, the bosonic susceptibility is the linear function of $\omega$, as indicated by the orange line. When temperature gets lower, the fermion goes into PG phase, prompting the bosonic scaling behavior to deriviate from linear function. And upon entering the SC phase, the $\chi^{-1} \sim \omega^2$ (the blue line, a guide to the eye) as in the FM phase. **c** Bosonic susceptibility in the disordered phase at $U = 8$ plotted for system size $L = 6, 8, 10, 12$ with various $\beta = 12, 16, 20, 24$. At high frequency all data points successfully merge together, as indicated by the red line quadratic in $\omega$ for $\omega > 1$. At low frequency, the $\chi^{-1} \sim \omega$ as indicated by the blue line, which is a guide to the eye, due to the non-conserved rotor order parameter.

From the analysis above, we see that the spin dynamics is consistent with the quantum-critical behavior and PG physics.

## Discussion

In this work, we performed a large-scale quantum Monte Carlo simulation of a FM spin-fermion model. We reported direct spectral and thermodynamic evidence of the formation of a PG prior to the SC transition. Within such a PG phase, the temperature evolution of the fermion spectral gap exhibits a gap-filling behavior, in sharp contrast with that of a conventional superconductor. Moreover, we found that the dynamics of the spin fluctuations display a different behavior than the well-known Landau damping behavior with $z = 3$.

Remarkably, we were able to reconcile all these features with theoretical predictions of Eliashberg theory and its generalization to the $\gamma$-model. Experimentally, PG phases have been observed in various unconventional superconductors[49,50], most notably the cuprates[51]. Our results imply that a PG arising from strong dynamical fluctuations should be ubiquitous in quantum-critical metals, and we expect this to be a fruitful direction for future research.

## Methods

**QMC simulations and data analysis**. We employ the determinant quantum Monte Carlo (DQMC) method[20,30] to simulate the Hamiltonian in Eq. (1). The quantum rotor model plays the role of the auxiliary field in the conventional DQMC and the quantum rotor model can be efficiently simulated with non-local update scheme developed in our previous work[13]. For each realization of the rotor in space-time, the fermion determinant is evaluated with the kinetic part and the coupling part of the Hamiltonian included as the configurational weight and the Markov chain of the Monte Carlo process is carried out according the weight. Detailed measurements of the physical observables are given in the Supplementary Note 2.

In order to obtain real-frequency spectral functions, the SAC scheme is employed to obtain the spectral function $N(\omega)$ from the imaginary-time correlation function $G(\tau)$,

$$G(\tau) = \int_{-\infty}^{\infty} d\omega \frac{e^{-\omega(\tau - \beta/2)}}{2\cosh(\beta\omega/2)} N(\omega) \qquad (3)$$

It is known that the problem of inverting the Laplace transform is equivalent to find the most probable spectra $N(\omega)$ out of its exponentially many suggestions to match the QMC correlation function $G(\tau)$ with respect to its stochastic errors, and such transformation has been converted to a Monte Carlo sampling process[52–54]. This QMC-SAC approach has been successfully applied to quantum magnets and interacting fermion systems ranging from the simple square lattice Heisenberg antiferromagnet[55] to deconfined quantum critical point and quantum spin liquids with their fractionalized excitations[56–58] and to the continuum model of twisted bilayer graphene and benchmarked with the exact solution at the chiral limit[59,60].

**Theoretical analysis**. We analyzed the QMC data for fermionic and bosonic response using the modified Eliashberg theory, which is a low energy effective dynamical theory for itinerant fermions near a QCP at finite temperatures. The theory accepts as parameters the static properties of a coupled fermion-boson system near a QCP, e.q. fermion bandstructure, bosonic susceptibility, etc., and computes the dynamical response of the system in terms of the fermionic self energy $\Sigma(\mathbf{k}, \omega_n)$ and bosonic self energy (polarization) $\Pi(\mathbf{q}, \Omega_n)$, taking into account the low energy excitations near the FS. It accounts for deviations from the canonical $T \to 0$ quantum critical behavior, e.g., deviations from the $\Sigma \sim \omega_n^{2/3}$ nFL self energy, and from the Landau damping $\Pi \sim \Omega_n/(v_F|\mathbf{q}|)$ as discussed in the main text. For details on the method see refs.[31,34].

We applied the theory to our QMC data, both to verify our assumptions on the normal state of the system and to extract the effective fermion-boson coupling. In the bare theory, the coupling $\bar{g} \sim K^2$, but it is renormalized by fermions with energies of order of the bandwidth, so it should be extracted by fitting from the QMC data. We present results for $U = 6$ which is almost above the QCP in Supplementary Fig. 13, showing good agreement between theory and data. For details of the fitting procedure and a discussion of the quality of the fits are presented in Supplementary Note 4. We found $\bar{g} = 6.3 \pm 0.2$, representing about a 20% renormalization of the bare vertex $K$, which is consistent with earlier works[34].

Finally, we used the obtained $\bar{g}$ to predict $T_{PG}$ within Eliashberg theory (the $\gamma$-model). Our model corresponds to $\gamma = 1/3$. The analytical prediction for $T_{PG}$ can be found in ref.[6], and details of the conversion from our $\bar{g}$ to the $\gamma$-model

parameters are in the Supplementary Note 4. We found $T_{PG} \approx 0.08$, in good agreement with the QMC $T_{PG} \sim 0.1$.

## Data availability

The data that support the findings of this study are available from the corresponding author upon reasonable request.

## Code availability

All numerical codes in this paper are available upon reasonable request to the authors.

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

## Acknowledgements

We thank R.M. Fernandes, M.H. Christensen, Y. Schattner, E. Berg, and X. Wang for valuable discussions. W.L.J. thanks Z. Liu for the support of the code, and G. Pan for the helpful suggestions. W.L.J., Y.Z.L. and Z.Y.M. acknowledge support from the RGC of Hong Kong SAR of China (Grant Nos. 17303019, 17301420, 17301721 and AoE/P-701/20), the Strategic Priority Research Program of the Chinese Academy of Sciences (Grant No. XDB33000000), the K. C. Wong Education Foundation (Grant No. GJTD-2020-01) and the Seed Funding "Quantum-Inspired explainable-AI" at the HKU-TCL Joint Research Centre for Artificial Intelligence. We thank the Center for Quantum Simulation Sciences in the Institute of Physics, Chinese Academy of Sciences, the Computational Initiative at the Faculty of Science and the Information Technology Services at the University of Hong Kong and the Tianhe platforms at the National Supercomputer Centers in Tianjin and Guangzhou for their technical support and generous allocation of CPU time. The authors also acknowledge Beijng PARATERA Tech CO.,Ltd.(https://www.paratera.com/) for providing HPC resources that have contributed to the research results reported within this paper. The work by A.V.C. was supported by the Office of Basic Energy Sciences, U.S. Department of Energy, under award DE-SC0014402. A.K. and A.V.C. acknowledge the hospitality of KITP at UCSB, where part of the work has been conducted. The research at KITP is supported by the National Science Foundation under Grant No. NSF PHY-1748958. Y.W. is supported by startup funds at the University of Florida and by NSF under award number DMR-2045871.

## Author contributions

A.K., Y.W., K.S., A.V.C. and Z.Y.M. initiated the work. W.L.J. and Y.Z.L. developed the program and performed the QMC calculations, W.L.J., Y.Z.L. and Z.Y.M. carried out the numerical data analysis. A.K., Y.W., K.S. and A.V.C. performed the theory analysis. All authors wrote the manuscript together.

## Competing interests

The authors declare no competing interests.
