## [Peer Review File · Nature Communications]

REVIEWER COMMENTS

Reviewer #1 (Remarks to the Author):

The paper by Jiang et al. studies a model of ferromagnetic criticality in a metal using sign-problem free quantum Monte-Carlo simulations. The results are not entirely unexpected, but it is good to establish them in a concrete model using reliable methods nevertheless. I think the manuscript can be published in Nature Communications after the authors have addressed the following questions:

- 1) I couldn't locate results for the diamagnetic susceptibility in the manuscript. Can the authors comment on if they observe onset of strong diamagnetic fluctuations in the pseudogap regime itself, before the onset of true SC?
- 2) There has been a great deal of past discussion (by Belitz, Kirkpatrick and others) on the fate of ferromagnetic criticality in clean metals being first-order in nature. While the data here indicates otherwise, I think there should at least be a brief discussion of this and the author's comments on the status of the theoretical work.

Reviewer #2 (Remarks to the Author):

The manuscript at hand reports the detailed numerical analysis of a spin-fermion model. The latter is typically formulated to capture the interplay of fluctuating gapless bosonic (magnetic) and fermionic (electronic) modes in the vicinity of a quantum critical point where the system loses its magnetic order. The motivation is to shed light on the origin of superconductivity and some of its more subtle acquaintances — the formation of a pseudogap phase above a superconducting dome and widespread anomalous transport behavior manifesting a non-Fermi liquid (nFL) regime.

On a technical level, the authors have constructed a two-flavor electronic model with $O(2)$ magnetic degrees of freedom driven through a quantum phase transition from a ferromagnetic to a magnetically disordered phase. This construction goes back to a seminal 2012 Science paper by Berg, Metlitski & Sachdev showing that such models are devoid of a fermion sign problem and therefore amenable to quantum Monte Carlo simulations. Unfortunately, this important reference is not mentioned in the manuscript at hand.

Finite-temperature transitions of such a two-dimensional model with $O(2)$ symmetry are generically Kosterlitz-Thouless (KT) transitions, i.e. true finite-temperature transitions. This allows to map out a phase diagram like the one in Fig. 1(b) which has several interesting features such as competing magnetic and electronic/superconducting orders including a "back bending" of the two phases upon approaching one another at finite temperature (resulting in a much emphasized reentrant magnetic behavior). Many of these features have first been reported for the $O(2)$ model with *antiferromagnetic* couplings in Ref. [17] of the current manuscript and a subsequent more detailed analysis in Ref. [18]. Again, the authors mention these striking similarities only in passing, but prefer to state that most earlier studies have preferred to study Ising magnetic degrees of freedom (which might be correct, but not to the point).

The most striking new data presented in the manuscript at hand are those showing the density of states in Fig. 2(a). This is really beautiful data which makes the case for the formation of a pseudogap (clearly visible in the low-temperature data) and the phenomenon of "gap filling". I certainly enjoyed looking at this data and its discussion. Technically, it is obtained by a (stochastic) analytical continuation, which unfortunately remains an uncontrolled technique. As such the statement by the authors that they report the "first unambiguous numerical realization of the transition from nFL to pseudogap and

eventually superconductivity" feels like an overstatement, particularly with regard to claim of an "unambiguous" result.

But one should nevertheless appreciate the reported observation of a pseudogap regime as such. It again follows in the footsteps of Ref. [17] where for the AFM XY model a regime of "diamagnetic fluctuations" was reported (as a stand-in for the pseudogap phase).

Comparing the phase diagrams reported there and in the manuscript at hand, one finds a very similar phenomenology – the thermal transition line into the pseudogap phase tracks the lower temperature transition into the SC phase, remaining roughly proportional for a wide regime. But is this in good agreement with experiments as the authors claim?

In summary, I enjoyed reading this manuscript and see the analysis of spin-fermion models being further expanded. The direct observation of the pseudogap in Fig. 2(a) [though obtained via analytical continuation] is a very nice result and an important point of reference in the future. This also holds for the subsequent discussion of the gap filling phenomenon. But I am left skeptical whether the manuscript in its current form should make the cut for publication in Nature Communications. For sure, the wording needs to be toned down when it comes to claims of novelty (as mentioned above), referencing of seminal work by others, and in comparing to the AFM spin-fermion model.

Response to Referee A

We thank the referee for careful reading of our manuscript (MS), positive assessment of our work and the recommendation to accept our MS for publication.

Comment 1: - I couldn't locate results for the diamagnetic susceptibility in the manuscript. Can the authors comment on if they observe onset of strong diamagnetic fluctuations in the pseudogap regime itself, before the onset of true SC?

Reply 1: Yes, we did observe strong diamagnetic fluctuations in the pseudogap region. In a lattice model, diamagnetic fluctuations are identified by computing the superfluid density ρ_s and identifying the temperature T_{dia} , below which ρ_s becomes positive. [In the thermodynamic limit $\rho_s = 0$ at $T \geq T_c$, but in a finite system there is a remnant signal, which allows one to detect the sign change of ρ_s above T_c]. We depict $\rho_s(T)$ in Fig. 1. The sign change is clearly observed near the upper edge of the pseudogap regime, marked in grey. We also show in Fig. 2 the scan of ρ_s vs U for a given $T > T_c$, which shows that ρ_s is positive above T_c in a wide range of U .

We added Figs. S8 and S9 to the Supplementary Information, reproduced as Figs. 1 and 2 below, where we also discussed how ρ_s as been detected in our numerics. We also marked T_{dia} line in the phase diagram in the Supplementary Information (see Fig. S10 in the Supplemental Information and Fig. 3 below.)

Figure 1: Superfluid density versus temperature at $U = 6.0$.

Figure 2: Superfluid density versus U at $\beta = T^{-1} = 16$.

Comment 2: There has been a great deal of past discussion (by Belitz, Kirkpatrick and others) on the fate of ferromagnetic criticality in clean metals being first-order in nature. While the data here indicates otherwise, I think there should at least be a brief discussion of this and the author's comments on the status of the theoretical work.

Reply 2: We agree. We added the discussion of this point to the main text, and also added several references to relevant works and reviews (Refs. 33,41,43,44 in the revised MS).

Response to Referee B

We would like to thank referee B for the positive evaluation of our work.

Comment 1: - On a technical level, the authors have constructed a two-flavor electronic model with

Figure 3: Phase diagram showing the boundary of the onset of the diamagnetic fluctuating and pseudogap phase.

O(2) magnetic degrees of freedom driven through a quantum phase transition from a ferromagnetic to a magnetically disordered phase. This construction goes back to a seminal 2012 Science paper by Berg, Metlitski & Sachdev showing that such models are devoid of a fermion sign problem and therefore amenable to quantum Monte Carlo simulations. Unfortunately, this important reference is not mentioned in the manuscript at hand.

Reply 1: We apologize for the oversight. Indeed, the 2012 paper by Berg et al was highly influential for sign-problem-free QMC. We cited this work (Ref. 14 in the revised MS).

Comment 2: - Finite-temperature transitions of such a two-dimensional model with O(2) symmetry are generically Kosterlitz-Thouless (KT) transitions, i.e. true finite-temperature transitions. This allows to map out a phase diagram like the one in Fig. 1(b) which has several interesting features such as competing magnetic and electronic/superconducting orders including a “back bending” of the two phases upon approaching one another at finite temperature (resulting in a much emphasized reentrant magnetic behavior). Many of these feature have first been reported for the O(2) model with *antiferromagnetic* couplings in Ref. [17] of the current manuscript and a subsequent more detailed analysis in Ref. [18]. Again, the authors mention these striking similarities only in passing, but prefer to state that most earlier studies have preferred to study Ising magnetic degrees of freedom (which might be correct, but not to the point).

Reply 2: We again apologize for the oversight. Indeed, there are various similarities between our results and those for antiferromagnetic O(2) model, including back-bending behavior and diamagnetic response above T_c . We explicitly write about this in the revised MS. As the referee pointed out, our key novel results are the direct observation of pseudogap behavior in the density of states (Fig. 2a)

and of the gap filling phenomena.

As we wrote in the reply to the first referee, we added the discussion on the diamagnetic response and stated explicitly that the same behavior has been observed in earlier QMC studies near an antiferromagnetic quantum critical point.

Comment 3: - The most striking new data presented in the manuscript at hand are those showing the density of states in Fig. 2(a). This is really beautiful data which makes the case for the formation of a pseudogap (clearly visible in the low-temperature data) and the phenomenon of “gap filling”. I certainly enjoyed looking at this data and its discussion. Technically, it is obtained by a (stochastic) analytical continuation, which unfortunately remains an uncontrolled technique. As such the statement by the authors that they report the “first unambiguous numerical realization of the transition from nFL to pseudogap and eventually superconductivity” feels like an overstatement, particularly with regard to claim of an “unambiguous” result.

Reply 3: We thank the referee for positive evaluation of our results and kind words. We followed the referee’s advice and toned down the wording, for example, to “an attempt to numerically realize the transition from nFL to pseudogap and eventually to superconductivity, ...”.

For the record, we present below our reasoning for the effectiveness of our analytical continuation. The QMC-SAC method for extracting the real-frequency spectra, which was developed over the past decades has been verified in many works on quantum magnetic and interacting fermion systems to be a reliable scheme to quantitatively obtain spectral information. For example, it has been directly compared with the Bethe ansatz, exact diagonalization, field theoretical analysis and even experiments, such as the works on 1D Heisenberg chain [Phys. Rev. E 94, 063308 (2016)], 2D Heisenberg model [Phys. Rev. X 7, 041072 (2017), Phys. Rev. Lett. 126, 227201(2021)], Z_2 quantum spin liquid model with fractionalized spectra [Phys. Rev. Lett. 121, 077201 (2018), npj Quantum Materials 6, 39 (2021)] and quantum Ising model with direct comparison with neutron scattering and NMR experiments [Nature Communications 11, 5631 (2020) and Nature Communications 11, 1111 (2020)]. More recently, it has been used to obtain the single-particle spectra and density of states in the continuum model of twisted bilayer graphene and benchmarked with the exact solution at the chiral limit [Chin. Phys. Lett. 38, 077305 (2021) and arXiv:2108.12559]. In these works, the method was able to extract clear and consistent features of the spectral functions. We therefore believe that, although uncontrolled in principle, the QMC-SAC scheme has been shown to provide reliable spectral information for many different systems. We added these arguments to the “Methods” section of the revised manuscript.

Comment 4: - But one should nevertheless appreciate the reported observation of a pseudogap regime as such. It again follows in the footsteps of Ref. [17] where for the AFM XY model a regime of “diamagnetic fluctuations” was reported (as a stand-in for the pseudogap phase). Comparing the phase diagrams reported there and in the manuscript at hand, one finds a very similar phenomenology — the thermal transition line into the pseudogap phase tracks the lower temperature transition into the SC phase, remaining roughly proportional for a wide regime. But is this in good agreement with experiments as the authors claim?

Reply 4: To a certain extent, yes. We mean, it has been argued by several scientists that the pseudogap region in the cuprates can be split into two parts: the “strong pseudogap regime” between T_c and T_p (with T_p roughly following T_c), where superconducting fluctuations dominate, and weak pseudogap regime at higher T , where observed pseudogap features are likely due to thermal magnetic fluctuations and/or precursors to Mott behavior. In this regard that clear gap filling behavior has

been observed only in the strong pseudogap region. We added a phrase on this and added relevant references.

The question about the relation between strong diamagnetic fluctuations (similar to that identified in Ref. 17) and the gap-filling pseudogap was also asked by the Referee 1. We refer to our reply to Referee 1 and to Figs. 1-3.

Comment 5: - In summary, I enjoyed reading this manuscript and see the analysis of spin-fermion models being further expanded. The direct observation of the pseudogap in Fig. 2(a) [though obtained via analytical continuation] is a very nice result and an important point of reference in the future. This also holds for the subsequent discussion of the gap filling phenomenon. But I am left skeptical whether the manuscript in its current form should make the cut for publication in Nature Communications. For sure, the wording needs to be toned down when it comes to claims of novelty (as mentioned above), referencing of seminal work by others, and in comparing to the AFM spin-fermion model.

Reply 5: Thanks again to the referee. In the revised MS we have followed all his/her recommendations.

List of changes in the revised manuscript

Below we list the changes in the revised manuscript. The main changes to the text are marked in red in the manuscript.

1. In response to the comments of Referee A, we added a discussion on previous works on ferromagnetic criticality in clean metals in the revised MS. We also amended the MS and added new data and a new section in the revised Supplementary Information, where we discussed the observed enhanced diamagnetic fluctuations inside the pseudogap temperature region.
2. In response to the comment of Referee B, we added the missing important reference such as the groundbreaking work on QMC in spin-fermion models (Ref. 14 in the revised MS). We compared and discussed our results with earlier important results on antiferromagnetic spin-fermion models.
3. In response to the comment of Referee B, we have toned down the wordings due to the limitation of the numerical method, and further discussed the reliability of the QMC-SAC scheme in terms of obtaining controlled spectral information, in the revised Method section.

REVIEWER COMMENTS

Reviewer #3 (Remarks to the Author):

In this manuscript the authors explore the development of a pseudogap (PG) in the density of states (DOS) of a toy-model of itinerant fermions constrained to move in two separated layers which are ferromagnetically coupled to SO(2) quantum rotors located in a middle layer. The authors perform a quantum Monte Carlo (QMC) study of the model obtaining its phase diagram. They fix the value of the spin-fermion interaction K to a rather large value, and they study the evolution of the system as a function of U/t_b , i.e., the parameters of the quantum rotors' Hamiltonian which is known to have a quantum critical point (QCP) at $(U/t_C)_c=4.25$. The authors then investigate how the position of the QCP is affected by the interaction with the fermions and how this interaction, in turn, affects the properties of the, otherwise, free fermions. They observed an exotic superconducting (SC) state, a spin triplet with S-wave symmetry. The study is performed at half-filling.

At very low temperature, they observe a ferromagnetic phase which eventually gives way to a superconducting state after U/t_b goes through a QCP, at a larger value than for the pure rotor model, around which the two phases coexist. The superconducting state is studied and they find interlayer S-wave pairs with spin 1, not surprising due to the tendency towards FM in the system. Using analytical continuation approaches the authors study the DOS. They observe the superconducting gap which increasing the temperature turns into a pseudogap that remains until a finite temperature T_{PG} . The authors argue that the PG results from the SC quantum fluctuations around the QCP and claim that this is the same mechanism that induces the PG observed in high T_c cuprate superconductors.

While the numerical results are interesting, I believe that it is a stretch to imply that there is a relationship with the mechanism that produces high T_c superconductivity in the cuprates, since there are very few commonalities between the two systems (the model studied and the cuprates) such as: i) Ferromagnetic (FM) as opposed to antiferromagnetic (AF) parent state; while there are indications that AF fluctuations induce exotic superconductivity, as observed in cuprates, pnictides, and heavy fermions, no similar results have been observed in FM systems; ii) doping is expected to play an important role while the current phenomenology is observed at half-filling taking advantage of the presence of two layers which could be identified with two different orbitals; the resulting pairing is interlayer, i.e., interorbital which does not appear to occur even in multiorbital systems such as the pnictides due to the on-site Coulomb repulsion; iii) the authors focus on the parameter U/t_b and only in passing discuss the value of the coupling K , just indicating that it is set at a larger value than in previous works whose citation is not provided.

Clearly the authors choose their toy model in order to avoid sign problems in their QMC simulations; while the results are interesting, the development of a pseudogap has already been numerically observed in various models for realistic materials, including cuprates, in which two phases compete. Thus, in my opinion, the results presented are not novel and the relationship of the model to high T_c cuprates is tenuous and thus, I do not recommend publication of this paper in Nature Communications.

In addition, there are various issues that the authors should address before the paper is ready to be published in any journal:

1) In Fig.3a the authors present the Fermi surface of the non-interacting system where the FS of the two half-filled layers are identical; in panel (b) the system is in its ferromagnetic phase and two FS can be seen; one would expect that the largest one corresponds to the

majority spins aligned parallel to the FM rotors while the smaller one corresponds to the fermions with anti-aligned spins. Could the authors indicate the layer composition of each FS? Finally, in panel (c) the authors show the FS in the low temperature paramagnetic phase of the interacting system. Could the authors say if the two bands are still degenerate or whether the degeneracy has been broken? This is not clear in the figure. Does the interaction with the rotors hybridize the two layers?

2) The authors attribute the reentrant behavior of the FM as an effect driven by SC fluctuations ruling out the possibility that the reentrant behavior of the SC phase is driven by the magnetic fluctuations. Since it has been speculated that AF fluctuations in doped cuprates could induce SC, shouldn't their result indicate that their model does not capture the physics of the cuprates? The authors should discuss this point in more detail.

3) If high T_c exotic D-wave superconductivity in the cuprates is expected to arise from antiferromagnetic fluctuations, why do the authors claim that their proposed mechanism which arises from a FM state affected by incoherent S-wave spin-triplet pairing correlations could apply to the cuprates? This is particularly puzzling because the authors claim that the pseudogap behavior has not been observed in previous studies of the same model but in its AF version. Do the authors believe that the previous calculations were incorrect or is the physics indeed different?

4) While the behavior reported is interesting, it is not clear that the model, with a FM and an exotic S-wave spin triplet SC state, could be realized in a real material. The interest of the paper would be enhanced if candidate materials could be presented. Notice that from the point of view of a multiorbital system the proposed pairing is "on-site" and thus, it would be very unlikely in a real material due to Coulomb repulsion.

Reviewer #4 (Remarks to the Author):

In this manuscript, the authors put forward a lattice model that could support both SC dome and pseudogap phase near a ferromagnetic QCP. The coupling between itinerant fermions and magnetic fluctuation around its critical point have been extensively studied by some of the authors using analytical methods. This work provides a concrete lattice model that can be investigated numerically. It could thus test some important results from previous studies, like spin-fluctuation induced superconductivity and the SC fluctuation induced pseudogap phase.

Also, the paper is clearly written, and the authors provide solid evidence to support their claims. Therefore, I would recommend it to be published in Nature Communication. I have no other issues with this revised version, except some suggestions that the authors may consider.

1. (Line 185, Page 3) The authors investigated various pairing channels and established the

dominant one as the layer-singlet s-wave pairing. I would suggest the authors provide more details in supplemental material, such as which pairing channels they consider and the data of correlation function for different channels. Readers interested in the computation details may find these to be useful.

2. (Line 212, Page 4) The authors mention fermionic spectral function, but only show the local DOS plot. If no further information could be extracted from spectral function, it would be better to rephrase this sentence to avoid confusion.

3. I suggest the authors add some data below T_c , either in Fig.2a or an individual plot in supplemental material. This could clearly demonstrate the difference between SC and pseudogap phase in the evolution of fermionic gap with temperature.

Reply to reviewer #3

Comment 1: - In this manuscript the authors explore the development of a pseudogap (PG) in the density of states (DOS) of a toy-model of itinerant fermions constrained to move in two separated layers which are ferromagnetically coupled to SO(2) quantum rotors located in a middle layer. The authors perform a quantum Monte Carlo (QMC) study of the model obtaining its phase diagram. They fix the value of the spin-fermion interaction K to a rather large value, and they study the evolution of the system as a function of U/t_b , i.e., the parameters of the quantum rotors' Hamiltonian which is known to have a quantum critical point (QCP) at $(U/t_b)_c = 4.25$. The authors then investigate how the position of the QCP is affected by the interaction with the fermions and how this interaction, in turn, affects the properties of the, otherwise, free fermions. They observed an exotic superconducting (SC) state, a spin triplet with S-wave symmetry. The study is performed at half-filling.

Reply 1: We thank the Referee for the nice summary of our model. However, we would like to point out that our work is not performed at half-filling (in fact, the word “half-filling” does not appear throughout the whole manuscript). None of our observed results crucially depends on a specific filling factor. Rather, we have two layers of fermions, but the doubling of electronic degrees of freedom do not give rise to any of the effects associated with half filling, e.g. nesting or Mott localization.

Comment 2: - At very low temperature, they observe a ferromagnetic phase which eventually gives way to a superconducting state after U/t_b goes through a QCP, at a larger value than for the pure rotor model, around which the two phases coexist. The superconducting state is studied and they find interlayer S-wave pairs with spin 1, not surprising due to the tendency towards FM in the system. Using analytical continuation approaches the authors study the DOS. They observe the superconducting gap which increasing the temperature turns into a pseudogap that remains until a finite temperature T_{PG} . The authors argue that the PG results from the SC quantum fluctuations around the QCP and claim that this is the same mechanism that induces the PG observed in high T_c cuprate superconductors. While the numerical results are interesting, I believe that it is a stretch to imply that there is a relationship with the mechanism that produces high T_c superconductivity in the cuprates, since there are very few commonalities between the two systems (the model studied and the cuprates) such as: i) Ferromagnetic (FM) as opposed to antiferromagnetic (AF) parent state; while there are indications that AF fluctuations induce exotic superconductivity, as observed in cuprates, pnictides, and heavy fermions, no similar results have been observed in FM systems; ii) doping is expected to play an important role while the current phenomenology is observed at half-filling taking advantage of the presence of two layers which could be identified with two different orbitals; the resulting pairing is interlayer, i.e., interorbital which does not appear to occur even in multiorbital systems such as the pnictides due to the on-site Coulomb repulsion;

Reply 2: We would like to clarify that we do not claim that our model directly describes a cuprate-like system. What we demonstrated in this MS using a convenient numerical platform is that pairing fluctuations near a quantum critical point of a metal can lead to pseudogap (PG) physics. We claim this is a universal feature of quantum-critical pairing, that has not been seen in a numerically-exact calculations. This mechanism, underlying the appearance of a pseudogap is quite universal and should appear in models with antiferromagnetic spin fluctuations, which are more relevant to the cuprate systems. This is indeed one of our future directions, but beyond the scope of the current work. We emphasize that PG-like behavior has been found experimentally in many materials, not just the cuprates. For example, it has been observed in the Se-doped FeSe, and in magic angle twisted bilayer graphene (e.g. Nat. Comm. 7, 12843 (2016), Nature 600, 240–245 (2021)). PG regimes

in these materials appear in the vicinity of a variety of electronic orders, including ferromagnetic, antiferromagnetic, valley, and charge-nematic orders.

Comment 3: - The authors focus on the parameter U/t_b and only in passing discuss the value of the coupling K , just indicating that it is set at a larger value than in previous works whose citation is not provided.

Reply 3: We thank the referee for pointing this out. Within our model, U/t_b controls the distance from the quantum-critical point, while K tunes the strength of pairing instability. To be able to analyze the pairing order and the corresponding pseudogap region, K has to be large enough as one varies the value of U/t_b . This is the crucial difference between earlier works with a smaller K . We did explicitly cite the previous works in a paragraph devoted to the comparison with earlier works (L81 in the previous manuscript). However we agree it is not clear so we added discussion.

Comment 4: - Clearly the authors choose their toy model in order to avoid sign problems in their QMC simulations; while the results are interesting, the development of a pseudogap has already been numerically observed in various models for realistic materials, including cuprates, in which two phases compete. Thus, in my opinion, the results presented are not novel and the relationship of the model to high T_c cuprates is tenuous and thus, I do not recommend publication of this paper in Nature Communications.

Reply 4: We apologize for the omission of citations of previous numerical work on the PG physics. We have added citations to some earlier works, including a review article (Phys. Rev. Lett. 110, 216405 (2013); Phys. Rev. X 5, 041041 (2016); Phys. Rev. X 8, 021048 (2018)) Indeed, in a broader context, it is well-known that PG behavior can be of various origin. We fully agree with the referee that competing orders can lead to a pseudogap, as has been discussed in various works. The scenario investigated here is one in which the PG behavior emerges from pairing fluctuations. It also has been extensively discussed, but as we said, it has not been studied in detail within the unbiased Monte-Carlo approach.

Comment 5: - In addition, there are various issues that the authors should address before the paper is ready to be published in any journal: 1) In Fig.3a the authors present the Fermi surface of the non-interacting system where the FS of the two half-filled layers are identical; in panel (b) the system is in its ferromagnetic phase and two FS can be seen; one would expect that the largest one corresponds to the majority spins aligned parallel to the FM rotors while the smaller one corresponds to the fermions with anti-aligned spins. Could the authors indicate the layer composition of each FS? Finally, in panel (c) the authors show the FS in the low temperature paramagnetic phase of the interacting system. Could the authors say if the two bands are still degenerate or whether the degeneracy has been broken? This is not clear in the figure. Does the interaction with the rotors hybridize the two layers?

Reply 5: We fully agree with the referee that the spin degeneracy is lifted by the ferromagnetic order parameter and the Fermi surface is split according to the spin polarization. To answer the referee's question on the layer degree of freedom — anywhere in the normal state, they are not hybridized by interacting with the rotors. In particular in the ordered state, each of the spin-split Fermi surfaces remain degenerate between top-layer and bottom-layer fermions. The figure is obtained by averaging over both layers, but the result would be the same only probing one layer.

Comment 6: - 2) The authors attribute the reentrant behavior of the FM as an effect driven by SC fluctuations ruling out the possibility that the reentrant behavior of the SC phase is driven by the magnetic fluctuations. Since it has been speculated that AF fluctuations in doped cuprates

could induce SC, shouldn't their result indicate that their model does not capture the physics of the cuprates? The authors should discuss this point in more detail.

Reply 6: We think there is a misunderstanding here — in our results, the SC phase does not exhibit reentrance behavior (which would imply a “bending-back” of the phase boundary as temperature decreases). Regarding relations to the cuprates, we do not claim our present model directly describes the cuprates. However, as we said in our second answer above, we argue that results observed should be universal to quantum-critical pairing problems, whether the “pairing glue” is of ferromagnetic or antiferromagnetic nature. Indeed, reentrance behavior of an AF phase due to pairing fluctuations has been seen in earlier theoretical and analytical works with an spin-density-wave quantum-critical point, which has been discussed analytically in e.g. Phys. Rev. B 80, 035117 (2009) and numerically in e.g. Phys. Rev. Lett. 117, 097002 (2016). We added a clarifying sentence on this.

Comment 7: - 3) If high T_c exotic D-wave superconductivity in the cuprates is expected to arise from antiferromagnetic fluctuations, why do the authors claim that their proposed mechanism which arises from a FM state affected by incoherent S-wave spin-triplet pairing correlations could apply to the cuprates? This is particularly puzzling because the authors claim that the pseudogap behavior has not been observed in previous studies of the same model but in its AF version. Do the authors believe that the previous calculations were incorrect or is the physics indeed different?

Reply 7: First, we should clarify that we are by no means claiming the pairing symmetry in the cuprates are s-wave spin-triplet. As we mentioned above, we do not claim our microscopic model directly applies to the cuprates. We do not claim that the earlier calculations on the antiferromagnetic quantum-critical point are incorrect. To the contrary, it is our belief that similar pseudogap also exists there. Indeed, plenty of experimental evidence on pseudogap induced by pairing fluctuations has been known, e.g. Nature Physics 5, 217–221 (2009), Phys. Rev. Lett. 101, 137002 (2008), Phys. Rev. B 87, 060506(R) (2013). However, as we mentioned above, one needs to go to a large enough coupling K in order to see this behavior within quantum Monte Carlo simulations, which has yet been done for antiferromagnetic models. An additional subtlety for antiferromagnetic models is that there the pairing gap is concentrated at the hot spots, and one needs to have a large momentum space resolution. To avoid future confusion, we have modified the manuscript and clearly states that we do not consider our model as the one intended for the cuprates.

Comment 8: - 4) While the behavior reported is interesting, it is not clear that the model, with a FM and an exotic S-wave spin triplet SC state, could be realized in a real material. The interest of the paper would be enhanced if candidate materials could be presented. Notice that from the point of view of a multiorbital system the proposed pairing is “on-site” and thus, it would be very unlikely in a real material due to Coulomb repulsion.

Reply 8: On the interesting issue of material realization of our model, there is no reason why the pairing symmetry cannot be realized in a two-layer material. We would like to also point out that the pairing within our model is not on-site. Rather the pairing is between two layers, and thus the issue of Coulomb repulsion can be alleviated by a large separation of the two layers and/or the existence of a middle spin layer separating the two itinerant ones. However, proposing a material realization is not our focus. As we mentioned, instead we are trying to show a general phenomenon that are widely believed to be universal using the most convenient numerical platform.

Reply to reviewer #4

Comment 1: - In this manuscript, the authors put forward a lattice model that could support both SC dome and pseudogap phase near a ferromagnetic QCP. The coupling between itinerant fermions and magnetic fluctuation around its critical point have been extensively studied by some of the authors using analytical methods. This work provides a concrete lattice model that can be investigated numerically. It could thus test some important results from previous studies, like spin-fluctuation induced superconductivity and the SC fluctuation induced pseudogap phase. Also, the paper is clearly written, and the authors provide solid evidence to support their claims. Therefore, I would recommend it to be published in Nature Communication. I have no other issues with this revised version, except some suggestions that the authors may consider.

Reply 1: We thank the referee for the positive evaluation of our work.

Comment 2: - 1. (Line 185, Page 3) The authors investigated various pairing channels and established the dominant one as the layer-singlet s-wave pairing. I would suggest the authors provide more details in supplemental material, such as which pairing channels they consider and the data of correlation function for different channels. Readers interested in the computation details may find these to be useful.

Reply 2: Indeed we did study the pairing correlations in other channels. Following the suggestion by the referee, we presented these details in the new version of the Supplementary Information.

Comment 3: - 2. (Line 212, Page 4) The authors mention fermionic spectral function, but only show the local DOS plot. If no further information could be extracted from spectral function, it would be better to rephrase this sentence to avoid confusion.

Reply 3: To be precise, what we showed is the density of states, which is a momentum-integral over the fermionic spectral function. Since the Fermi surface is rather isotropic in momentum space, showing an k -dependent spectral function does not provide much additional information. We have amended the misleading sentence.

Comment 4: - 3. I suggest the authors add some data below T_c , either in Fig.2a or an individual plot in supplemental material. This could clearly demonstrate the difference between SC and pseudogap phase in the evolution of fermionic gap with temperature.

Reply 4: This is a very good point, and we have followed the referee's suggestion to add new spectral data with $T = 1/24$ and $1/30$ below T_c in Fig.2a of MS.

List of changes in the revised manuscript

Below we list the changes in the revised manuscript. The main changes to the text are marked in red in the manuscript.

1. Added a number of the references mentioned in the response.
2. Added several clarifying sentences to the introduction, and an additional final paragraph.
3. Modified several sentences in the body of the text to clarify some of the review points.
4. Added new data plots in the MS and Supplementary Information, as mentioned in the response.

REVIEWER COMMENTS

Reviewer #4 (Remarks to the Author):

I am confused with the authors' response considering pairing channels. I previously thought they considered a number of possible pairings and arrived at the conclusion that the spin-triplet one is dominant among those channels. But in the supplemental material, they only show the results for two channels. If the authors only considered the two channels, they should make it clear in their main text instead of just saying "measuring correlation functions of Cooper pairs in various pairing channels". Also, it would be better to explain why there is no need to look at other channels such as s-wave pairing in the same layer. If that is not the case, however, I would strongly suggest they provide numerical data of all the pairing channels they've already investigated.

Reply to reviewer #4

Comment 1: I am confused with the authors' response considering pairing channels. I previously thought they considered a number of possible pairings and arrived at the conclusion that the spin-triplet one is dominant among those channels. But in the supplemental material, they only show the results for two channels. If the authors only considered the two channels, they should make it clear in their main text instead of just saying "measuring correlation functions of Cooper pairs in various pairing channels". Also, it would be better to explain why there is no need to look at other channels such as s-wave pairing in the same layer. If that is not the case, however, I would strongly suggest they provide numerical data of all the pairing channels they've already investigated.

Reply 1: This is a valid question. We do have additional data which we now display in full in the Supplementary Information (SI). Specifically, we list 10 different pairing channels in Table 1 and show susceptibilities in all these channels in Figs. 1 and 2 of the SI as functions of the interaction U and the temperature. We clearly see that only the susceptibility for the s-wave layer singlet and spin-triplet ($S = 0$) order parameter (the one labeled as $C_{os,st0}$ in the Table) rapidly increases with increasing U and/or decreasing T . Susceptibilities for all other order parameters remain small and weakly change with U and T .

For convenience of the referee we copy Table 1 here:

Channel	Description	Definition
$C_{os,is}$	On-site, s -wave, Intra-layer, spin-singlet	$\frac{1}{\sqrt{2}}(\hat{c}_{i1\uparrow}\hat{c}_{i1\downarrow} + \hat{c}_{i1\downarrow}\hat{c}_{i1\uparrow})$
$C_{os,ts}$	On-site, s -wave, layer-triplet, spin-singlet	$\frac{1}{\sqrt{2}}(\hat{c}_{i1\uparrow}\hat{c}_{i2\downarrow} - \hat{c}_{i1\downarrow}\hat{c}_{i2\uparrow})$
$C_{os,st0}$	On-site, s -wave, layer-singlet, spin-triplet ($S = 0$)	$\frac{1}{\sqrt{2}}(\hat{c}_{i1\uparrow}\hat{c}_{i2\downarrow} + \hat{c}_{i1\downarrow}\hat{c}_{i2\uparrow})$
$C_{os,st1}$	On-site, s -wave, layer-singlet, spin-triplet ($S = 1$)	$\frac{1}{\sqrt{2}}(\hat{c}_{i1\uparrow}\hat{c}_{i2\uparrow} - \hat{c}_{i2\uparrow}\hat{c}_{i1\uparrow})$
$C_{ns,is}$	Nearest-neighbor, s -wave, Intra-layer, spin-singlet	$\frac{1}{\sqrt{8}} \sum_l f_{ns}(\delta_l)(\hat{c}_{i1\uparrow}\hat{c}_{i+\delta_l1\downarrow} + \hat{c}_{i1\downarrow}\hat{c}_{i+\delta_l1\uparrow})$
$C_{ns,ts}$	Nearest-neighbor, s -wave, layer-triplet, spin-singlet	$\frac{1}{\sqrt{8}} \sum_l f_{ns}(\delta_l)(\hat{c}_{i1\uparrow}\hat{c}_{i+\delta_l2\downarrow} - \hat{c}_{i1\downarrow}\hat{c}_{i+\delta_l2\uparrow})$
$C_{ns,st0}$	Nearest-neighbor, s -wave, layer-singlet, spin-triplet ($S = 0$)	$\frac{1}{\sqrt{8}} \sum_l f_{ns}(\delta_l)(\hat{c}_{i1\uparrow}\hat{c}_{i+\delta_l2\downarrow} + \hat{c}_{i1\downarrow}\hat{c}_{i+\delta_l2\uparrow})$
$C_{ns,st1}$	Nearest-neighbor, s -wave, layer-singlet, spin-triplet ($S = 1$)	$\frac{1}{\sqrt{8}} \sum_l f_{ns}(\delta_l)(\hat{c}_{i1\uparrow}\hat{c}_{i+\delta_l2\uparrow} - \hat{c}_{i1\uparrow}\hat{c}_{i+\delta_l2\uparrow})$
$C_{np,it0}$	Nearest-neighbor p_x -wave, Intra-layer, spin-triplet ($S = 0$)	$\frac{1}{\sqrt{4}} \sum_l f_{np}(\delta_l)(\hat{c}_{i1\uparrow}\hat{c}_{i+\delta_l1\downarrow} + \hat{c}_{i1\downarrow}\hat{c}_{i+\delta_l1\uparrow})$
$C_{np,it1}$	Nearest-neighbor p_x -wave, Intra-layer, spin-triplet ($S = 1$)	$\frac{1}{\sqrt{2}} \sum_l f_{np}(\delta_l)\hat{c}_{i1\uparrow}\hat{c}_{i+\delta_l1\uparrow}$

The labels 1 and 2 represent layers. The coefficient $f_{ns}(\delta_l) = 1$ for $\delta_l = \hat{x}, \hat{y}$, and $f_{ns}(\delta_l) = -1$ for $\delta_l = -\hat{x}, -\hat{y}$, while $f_{np}(\delta_l) = 1$ for $\delta_l = \hat{x}$, and $f_{ns}(\delta_l) = -1$ for $\delta_l = -\hat{x}$.

We hope that the referee finds these results convincing.

List of changes in the revised manuscript

Below we list the changes in the revised manuscript. The changes are marked by red in the manuscript.

1. We put the discussion of the 10 pairing channels with an on-site order parameter into the SI.
2. We added susceptibilities of the other pairing channels to Figures 1 and 2 in the SI
3. We modified some sentences in the SI.

REVIEWERS' COMMENTS

Reviewer #4 (Remarks to the Author):

The revised manuscript has addressed all my concerns, and I am happy to recommend it to be published in Nature Communication.

Reply to reviewer #4

Comment 1: The revised manuscript has addressed all my concerns, and I am happy to recommend it to be published in Nature Communication.

Reply 1: We thank the respected referee for the recommendation. In the final manuscript, we marked the revised part with red to meet the requirements of format.